# The Predominant Sources of Heavy Metals in Different Types of Fugitive Dust Determined by Principal Component Analysis (PCA) and Positive Matrix Factorization (PMF) Modeling in Southeast Hubei: A Typical Mining and Metallurgy Area in Central China

**DOI:** 10.3390/ijerph192013227

**Published:** 2022-10-14

**Authors:** Hongling Chen, Dandan Wu, Qiao Wang, Lihu Fang, Yanan Wang, Changlin Zhan, Jiaquan Zhang, Shici Zhang, Junji Cao, Shihua Qi, Shan Liu

**Affiliations:** 1School of Environmental Science and Engineering, Hubei Polytechnic University, Huangshi 435003, China; 2Hubei Key Laboratory of Mine Environmental Pollution Control and Remediation, Hubei Polytechnic University, Huangshi 435003, China; 3Research Center of Ecological Environment Restoration and Resources Comprehensive Utilization, The First Geological Brigade of Hubei Geological Bureau, Huangshi 435000, China; 4School of Environment and Health, Jianghan University, Wuhan 430056, China; 5Institute of Atmospheric Physics, Chinese Academy of Sciences, Beijing 100029, China; 6School of Environmental Studies, China University of Geosciences, Wuhan 430074, China

**Keywords:** heavy metals, fugitive dust, principal component analysis, positive matrix factorization, source identification

## Abstract

To develop accurate air pollution control policies, it is necessary to determine the sources of different types of fugitive dust in mining and metallurgy areas. A method integrating principal component analysis and a positive matrix factorization model was used to identify the potential sources of heavy metals (HMs) in five different types of fugitive dust. The results showed accumulation of Mn, Fe, and Cu can be caused by natural geological processes, which contributed 38.55% of HMs. The Ni and Co can be released from multiple transport pathways and accumulated through local deposition, which contributed 29.27%. Mining-related activities contributed 20.11% of the HMs and showed a relatively high accumulation of As, Sn, Zn, and Cr, while traffic-related emissions contributed the rest of the HMs and were responsible for the enrichment in Pb and Cd. The co-applied source-identification models improved the precision of the identification of sources, which revealed that the local geological background and mining-related activities were mainly responsible for the accumulation of HMs in the area. The findings can help the government develop targeted control strategies for HM dispersion efficiency.

## 1. Introduction

Fugitive dust is considered the main fraction of particulate matter and is generally classified as construction fugitive dust, soil fugitive dust, landfill fugitive dust, road fugitive dust, deposit fugitive dust, and so on [1,2,3]. Previous studies have already revealed that fugitive dust accounts for 12–34% of particulate matter in most Chinese cities [4]. Additionally, fugitive dust easily enriches harmful elements in the environment and can cause potential human health risks, as it is a kind of small particle with a large specific surface [5,6]. In Tianjin, Wang et al. revealed an unacceptable cancer risk from Cr from urban road fugitive dust mainly originating from traffic non-exhaust emissions [7]. Cui et al. found the Ni, Cr, As, and Cd in fugitive dust can cause potential health risks and mainly originate from industrial activities in Tangshan [8]. As these previous studies indicate, with high-speed development, there has been a focus on source analysis of heavy metal pollution in fugitive dust in Chinese cities in recent years [5,8]. Much research, especially in large or industrial cities, has discussed fugitive dust from different sources, such as motor vehicles, fuel burning, construction activities, industrial activities, agricultural activities, and so on, [9,10]. Unfortunately, fewer studies have been undertaken to distinguish the sources of heavy metals (HMs) in mining and metallurgy areas, where fugitive dust may bind with relatively high heavy metal contaminants [11].

With thousands of years of mining and metallurgy history, southeast Hubei is the largest polymetallic deposit in central China [12,13,14]. Due to the long-term extensive mining activities, former studies have revealed that HMs in local fugitive dust, such as Cu, Zn, As, Pb, Cd, and so on, can cause potential cancer risks in citizens [11,15]. Besides contributing to the understanding of the health risks from fugitive dust, local governments should be interested in finding out where the HMs in fugitive dust may be from to develop accurate atmospheric pollution control policies [16]. However, different types of fugitive dust may have different transport pathways, which can influence the source of HMs, and these pathways are usually complicated in mining and metallurgy areas [17,18]. In such cases, it is necessary to undertake source identification of the HMs in the different types of fugitive dust in the area.

Previous researchers have generally used geostatistical models, receptor models, and multivariate statistical analyses to identify HM sources [19,20,21]. Principal component analysis (PCA) has been widely used and is considered a good model for potential source distinction [6]. Anaman et al. [22] employed the PCA model to determine the sources and transport routes of heavy metals (HMs) in different land-use soils. Receptor models, such as the positive matrix factorization (PMF) model, are generally used to determine source contributions and the potential sources of contaminants when there is no prior information on source profiles [23,24]. By using PMF, Chai et al. found that HMs in cultivated soil might originate from natural, industrial, and agricultural activities, as well as from traffic emissions [25]. Although the PCA and PMF method can be used to identify potential sources, many researchers still believe that both models have their weaknesses [26,27]. In recent years, more and more research has focused on the co-application of PCA and PMF for source apportionment [28]. Researchers believe that co-applied PCA and PMF can improve the accuracy of the determination of the pollutant contributions from each source, increase the precision of potential source identification, and minimize unsuitable estimations in source apportionment [29]. Such a method could contribute to a better understanding of the potential sources of HM pollutants in fugitive dust in mining and metallurgy areas [30]. Additionally, the precision with which the sources and transport pathways of HMs in fugitive dust can be identified is the key point for the development of policies for precise prevention and control of pollutants in such areas by local governments [16]. However, such research has been undertaken less often.

Given the above, source identification of HMs based on the different types of fugitive dust is poorly understood. Very few researchers have gone further to undertake source identification with combined source analysis methods in mining and metallurgy areas. To fill this knowledge gap, in the present research, the pollution characteristics and sources of HMs in five types of fugitive dust were comprehensively analyzed with co-applied PCA and PMF in a mining and metallurgy area. The main objectives of the study were: (1) to reveal the HM pollution characteristics in five different types of fugitive dust; (2) to determine the sources by combining PCA and PMF models; and (3) to compare the differences in source identification methods. The achievements should help increase the understanding of the sources and contributions of HMs in fugitive dust and provide scientific support for making accurate atmospheric pollution control policies in the biggest mining and metallurgy area in southeast Hubei.

## 2. Materials and Methods

### 2.1. Study Area

The study area was located in central China (114°31′–115°30′ E and 29°30′–30°15′ N) and has a population of 2.4 million and a gross domestic product of 0.16 trillion yuan. The climate of the area is humid subtropical, with an average annual temperature of 17 °C and annual rainfall of 1382 mm. The area has the largest copper and iron reserves in central China [31]. With long-term mining and metallurgy activities, many researchers have already found that the soil, water, and air have been polluted by heavy metals, such as Cu, Fe, Pb, Zn, etc. [12,15].

### 2.2. Sampling and Preparation

After at least seven consecutive days without rain, a total of 40 fugitive dust samples were collected, including 8 construction fugitive dust (CD), soil fugitive dust (SD), landfill fugitive dust (LD), road fugitive dust (RD), and deposit fugitive dust (DD) samples, respectively. (Figure 1). The collection of all the samples was undertaken from 15 October to 20 October 2020. Detailed information on the sampling locations is shown in Figure 1 and Appendix A.

Based on the sampling location, approximately 100 g of each of the following five types of fugitive dust was collected: (1) construction fugitive dust, which was collected from construction sites; (2) soil fugitive dust, which was sampled from the surface of the soil; (3) landfill fugitive dust, which was collected from the surface of the tailings reservoir and waste dumping sites; (4) road fugitive dust, which was sampled from the main street; (5) deposit fugitive dust, which was collected from windowsills and roofs that had not been cleaned for a longtime. Additionally, deposit fugitive dust was collected at a height of 1.5–2.0 m, while other types of fugitive dust were sampled on impervious surfaces within a 5 m^2^ circle. During the collection, the fugitive dust was sampled using trays and plastic brushes in a gentle sweeping motion to accumulate fine particulates. After each collection, the sampling tools were cleaned with paper towels. All samples were stored in paper bags wrapped with solvent-rinsed aluminum foil and then sealed in polyethylene bags for transportation to the laboratory. During sampling, each collection point was marked with a GPS locator (Appendix A). After collection, samples were air-dried to get rid of moisture. Then, small stones and coarse debris were removed with a 100 µm sieve. Moreover, all samples were ground and homogenized with an agate mortar. After grinding, samples were sieved in a 63 μm sieve. Finally, all prepared samples were placed in an air-tight container for storage.

### 2.3. Chemical Analyses

Eleven heavy metals (As, Cd, Co, Cr, Cu, Mn, Ni, Pb, Zn, Fe, and Sn) were measured from all the fugitive dust samples. Following the procedure used by Liu et al. [6], each sieved sample (0.2 g) was digested with HCl-HNO_3_-HF-HCLO_4_. Then, an atomic fluorescence spectrophotometer (AFS, AFS-230E) and inductively coupled plasma source-mass spectrometer (Perkin–Elmer, Elan 9000) were used to measure all the chemical elements in the digested solution. For the analyses, guaranteed reagents with high purity grades were used to avoid contamination of the samples [12]. To control the quality, a national standard one-level soil sample (GSS-23) was also analyzed with the fugitive dust samples. The recovery rate of each element was between 91.2% and 108.2%. The results of the sample duplicates (10%), the national standard samples, and the method blanks showed that the relative double difference was less than 10%. Moreover, the concentration of the target heavy metal elements in the procedural blanks was below the limit of detection. All the test results conformed to monitoring requirements.

### 2.4. Principal Component Analysis

PCA is a multivariate statistical analysis method that can reflect most of the original multivariate information with fewer variables [32]. The method assumes that a smaller set of principal components will be comprised of linear combinations of the original variables [33]. Each principal component is uncorrelated [29,32]. Moreover, the mass of contaminants should not change during transportation [6]. Numerous previous studies have used PCA to identify the natural and anthropogenic contributions of contaminants by extracting the principal component information from the variables [34,35]. The first component explains the maximum amount of data variance. Then, each successive component explains the maximum amount of remaining unexplained data. In such cases, it creates an orthogonal distribution of components [36]. In the current research, PCA was employed to reveal the relationships between different HMs and to indicate contaminant sources. The dimensionless standardized equation was used, as follows (Equation (1)):(1)$$Z_{ij}=\frac{C_{ij}-\bar{C_{j}}}{\sigma_{j}}$$
where *Z_ij_* represents the standardized score, *i* represents samples, *j* represents components, *C_ij_* represents the concentration of component *j* in sample *i*; $\bar{C_{j}}$ represents the arithmetic mean concentration for component *j*; and $\sigma_{j}$ represents the standard deviation for component *j*.

### 2.5. Positive Matrix Factorization

The PMF model is a receptor model that has been widely used to analyze the potential sources of various components since the early 1990s [37,38]. The receptor model assumes that the speciated data re influenced by linear combinations of source emissions, which are distributed as factor contributions [28,39]. The main goal of the models is to solve the chemical mass balance between measured species concentrations and source profiles [32]. In this research, the PMF model was employed to apportion the potential sources of HMs in fugitive dust. Based on the PMF user guide [39], the following equation was used (Equation (2)):(2)$$X_{ij}=\sum_{k=1}^{p}g_{ik}f_{kj}+e_{ij}$$
where *X_ij_* represents the measurement content of the heavy metal, *p* represents pollutant sources, *f* represents the source profile species, *g* represents the factor of contribution, *i* represents samples, *j* represents pollutant species, and *e_ij_* represents residual for sample and pollutant species.

Moreover, the optimal profiles and contributions that minimize the objective function Q can be derived with PMF (Equation (3)). The equation is as follows:(3)$$Q=\sum_{i=1}^{n}\sum_{j=1}^{m}{\left(\frac{x_{ij}-\sum_{k=1}^{p}g_{ik}f_{kj}}{u_{ij}}\right)}^{2}=\sum_{i=1}^{n}\sum_{j=1}^{m}{\left(\frac{e_{ij}}{u_{ij}}\right)}^{2}$$
where *u* represents the uncertainty of the concentration, which can be calculated as follows (Equations (4) and (5)):(4)$$\;\mathrm{For}\;c\leq \mathrm{MDL},u_{ij}=5/6\times \mathrm{MDL}$$
(5)$$\;\mathrm{For}\;c\;\leq \;\mathrm{MDL},\;u_{ij}=\sqrt{{\left(\mathrm{error}\;\mathrm{fraction}\times c\right)}^{2}+{\mathrm{MDL}}^{2}}$$
where c represents the concentration of the chemical species, MDL represents the species-specific method detection limit, and the error fraction represents the relative standard deviation [30].

### 2.6. Data Analysis

The SPSS 22.0 statistical package (Statistical Product and Service Solutions, SPSS Inc., Chicago, IL, USA) was employed to calculate the descriptive data for the heavy metals, such as minimum, maximum, mean, and standard deviation (SD). The source analysis of the pollutants was completed using PCA (Statistical Product and Service Solutions, SPSS Inc., Chicago, IL, USA) and PMF 5.0 (Sonoma Technology Inc., Petaluma, CA, USA). Two data mapping software packages were also used, including ArcGIS Desktop 10.5 (ESRI, Redlands, CA, USA) and OriginPro 2019C (OriginLab, Northampton, MA, USA).

## 3. Results and Discussion

### 3.1. Statistical Description of Heavy Metals in Fugitive Dust

Table 1 shows the results for the HM content in fugitive dust. The concentrations of HMs in all fugitive dust samples were ranked as follows: Cd (2.36 ± 3.08) < Sn (7.88 ± 9.11) < Co (18.4 ± 7.56) < Ni (43.8 ± 22.5) < As (82.9 ± 69.9) < Cr (139 ± 89.5) < Pb (240 ± 258) < Zn (409 ± 450) < Cu (437 ± 565) < Mn (683 ± 367) < Fe (39,023 ± 18,553). Additionally, the coefficient of variation (CV) of the HMs was relatively high, more than 41.0%. The results indicated that the spatial distribution of HMs was heterogeneous in the local area [8]. Moreover, in all detected HMs, the CV values of Cd, Cu, Pb, Zn, and Sn were 130%, 129%, 107%, 110%, and 116%, respectively. As the study area is the largest mining and metallurgy area in central China with a long-term mining history and copper ore, tin ore, and lead-zinc ore are the main minerals in the area, the local mining and metallurgy activities, industrial activities, traffic-related activities, and so on likely caused such heterogeneous emissions of heavy metals [12,13]. Moreover, the CV values of most HMs, such as Cd, Cu, Mn, Pb, Zn, and Sn, exceeded 54.7% in the CD and LD. The results might be related to the heterogeneous distribution of construction sites and landfill sites in the area [18]. Additionally, the emission of HMs in such sites was generally influenced by the construction materials and landfill waste, which might have resulted in the difference HM concentrations from site to site [40]. The CV values of As, Co, Cr, Mn, and Fe in the DD and RD were lower than 48.5%. The results indicate that such HMs had relatively homogeneous distributions, which might have been caused by the local atmospheric deposition [41].

The results in Figure 2 show the average accumulation concentration (AC) versus the local background value (BV) of the heavy metals, which were ranked as follows: Cu (14.24) > Cd (13.73) > Pb (8.99) > As (6.74) > Zn (4.89) > Sn (3.58) > Co (1.20) > Cr (1.61) > Ni (1.17) > Fe (1.00) > Mn (0.96). The results reveal that, except for Fe and Mn, the heavy metals in the fugitive dust were affected by human activities, especially Cu, in which the AC value was 0.24 to 96.49 times higher than the BV. In comparison with other research, the average concentrations of Cu, Pb, Zn, Cd, Mn, and Fe were relatively higher than those in Beijing [42], Hangzhou [43], Guangzhou [44], Nanjing [45], Toronto [46], and Shiraz [47]. The results reveal that mining-related activities increase the concentrations of HMs in particulate matter [11]. As the study area is the largest copper mining area in central China [31], the accumulation of Cu in the fugitive dust may be related to mining activities [48,49,50]. Comparing the different types of fugitive dust in the area, Figure 2 shows that the AC/BV values for As, Cd, Co, Cr, Cu, Ni, Pb, Zn, and Sn in the SD, DD, RD, and LD were relatively higher than in the CD. Previous studies have already revealed that toxic metals can be amassed through human activities [10]. Al-Shidi et al. found that the relatively high concentrations of Pb and Cd in fugitive dust originated from traffic emissions [51]. Additionally, in the local mining area, heavy trucks are employed to transport minerals, which can also increase the accumulation of Pb and Cd in RD [41,52]. Tian et al. also found that the AC/BV values of Mo, Cd, and Pb were significantly higher than those of other HMs in road dust in the Bayan Obo mining region [16]. Additionally, a multitude of industrial smelting plants can easily cause local HM pollution and increase the accumulation of HMs, such as As, Cd, Cu, Ni, Zn, and Pb, in DD and SD [53,54]. Xu et al. found that the average concentrations of Cu, Cd, and Ni in surface soil were 4.7, 17.1, and 3.7 times higher than their BVs in the Yangxin mining area [31]. Moreover, mining and metallurgy activities always result in significant amounts of tailings and numerous waste dumps, which increase the accumulation of HMs in LD [17]. However, construction activity is more likely to be nonrecurring, indicating that it would be more difficult for HMs to amass in CD [40].

### 3.2. Correlations of Heavy Metals in Fugitive Dust

The previous studies have already revealed that Pearson correlation analysis can be used to preliminarily determine the relation between HM concentrations and efficiently determine different sources by analyzing the linear relationship between two different HMs [55,56]. Significant positive correlations between two different HMs generally suggest a closely related source, while weak positive correlations indicate a different origin [20]. Moreover, the results of Pearson correlation analysis can also be used to indicate that the factor analyses from PCA and PMF are reliable [25]. The correlation coefficient value (r) describes the degrees of linear correlation. When |r| > 0.3, it indicates that a correlation exists between the two HMs, while when |r| > 0.7, a relatively high correlation exists [56]. Figure 3 shows the significant positive correlations between Cr and Ni (r = 0.737), Cr and Zn (r = 0.813), Cr and Sn (r = 0.802), Cu and Mn (r = 0.793), Mn and Fe (r = 0.721), Ni and Sn (r = 0.772), and Zn and Sn (r = 0.783), implying that these HMs may be homologous [57]. Week correlations between As and Cd (r = 0.0199), As and Mn (r = 0.242), As and Pb (r = 0.156), As and Fe (r = 0.271), Cd and Cr (r = 0.159), Cd and Mn (r = 0.0259), Cd and Sn (r = 0.278), Co and Pb (r = 0.0262), Co and Zn (r = 0.252), Cu and Pb (r = 0.205), Mn and Ni (r = 0.266), Mn and Pb (r = 0.155), and Pb and Fe (r = 0.237) imply that these HMs may not be homologous [57]. Moreover, other HMs showed moderate correlations, implying that they may come from multiple sources [55].

In the study area, manganese is the most important mineral associated with copper-iron ore [31]. Moreover, Mn showed strong positive correlations with Cu and Fe. The results suggest that the accumulations of Cu, Mn, and Fe relate to the geological background [58]. Cd and Pb showed a weak correlation with most HMs, indicating that Cd and Pb might be affected by independent sources [57]. However, it should be noted that it is hard to determine the sources of HMs by only using correlation analysis [25]. Such analyses can only preliminarily determine the relations between HMs. Building on the results of correlation analysis, more work should be undertaken to determine the sources of HMs in fugitive dust by employing PCA and PMF models.

### 3.3. Source Identification with PCA

Researchers generally use PCA to determine the most essential factors and minimize data with minimal data loss [59]. Before the source identification, the Kaiser–Meyer–Olkin index was 0.759 and the Bartlett sphericity test result was below 0.001, indicating that the PCA model would be a suitable method for determining the principal components (PCs) of the sources of HM in fugitive dust in the study area [6]. Moreover, Liu et al. classified the degrees of positive loading with loading values as strong positive loading (0.75–1.0), moderate positive loading (0.5–0.75), and weak positive loading (0.3–0.5) [6]. A high loading value indicates that the HMs may be influenced by a similar source [20]. Appendix A shows three PCs extracted from HMs in fugitive dust based on varimax rotation. The cumulative contributions of these PCs could explain 78.11% of their variance in fugitive dust.

In Figure 4 and Appendix A, PC1 explained 49.62% of the total variance and had a strong loading for Mn, Cu, and Fe. The results revealed that these HMs might come from the same source or similar transport pathways [28]. The metallogenic mechanism and local geology background indicate that PC1 was affected by natural geochemical processes [14]. Only As showed a strong positive loading in PC2, which accounted for 18.53% of the total variance, while multiple HMs had moderate positive loading in PC2, such as Cr, Sn, Ni, Co, and Zn. The results indicated that PC2 might have been influenced by multiple sources [29]. Considering the developing characteristics of the study area, the accumulation of Cr, Sn, Ni, Co, and Zn might relate to mining and industrial activities [60]. PC3, which described 9.96% of the total variance, had strong positive loading for Pb and Cd. Previous researchers generally believe that accumulation of Pb and Cd can be caused by traffic activities [29]. Additionally, the concentrations of HMs which were explained by PC2 and PC3 were significantly higher than the local background value, indicating that such HMs might be affected by anthropogenic activities [23]. Moreover, all the results analyzed by the PCA model showed good agreement with those of the correlation analysis. However, all the results from the PCA showed that the source of PC2 could not be accurately identified. A previous study has already revealed that it is sometimes difficult to analyze the characteristic markers for mixed sources with PCA [20]. Hence, quantitative source apportionment was further analyzed using PMF.

### 3.4. Quantitative Source Apportionment with PMF

According to the results of the correlation analysis and the PCA, there is no doubt that the HMs in fugitive dust in the mining and metallurgy area originate from nature and mining-related activities. To reveal quantitative information about the contributions from different sources, the PMF model was used to identify the major sources of HMs. Different factor numbers, such as 3, 4, and 5, were set for 20 runs in the PMF model to find high R^2^ and minimum Q values [61,62]. The Q value was the lowest and the residual value was in the range of −3 to 3 when the factor number was 4. The results indicate that this solution could explain the original data well [63]. Consequently, in Figure 5, four main factors, natural source (38.55%), local deposition (29.27%), mining-related activities (20.11%), and traffic-related source (12.06%), could fully explain the source information for the fugitive dust.

In Appendix A, the first factor was identified as natural sources, which was distinguished by relatively high contributions for Cu (77.6%), Mn (71.8%), and Fe (38.6%). The study area is one of the largest skarn deposits in China. The main mineral species include covellite, chalcopyrite, pyrite, and pyrrhotite [64], indicating that Cu and Fe exist widely in local geological processes. Furthermore, only the average concentrations of Mn and Fe were below the local background values (Table 1). Most researchers consider Mn and Fe to be the main elements of the Earth’s crust [65,66], indicating that they might originate from the geological weathering process.

For the second factor in Appendix A, the results showed relatively high loading values for Ni and Co, accounting for 41.1% and 45.7%, respectively. Compared with the loading values of Ni and Co in the PCA model, as the results indicated weak to moderate positive loading in multiple principal components, which indicates that these HMs have multiple transport pathways [20,36]. Guan et al. has already revealed that Ni is the marker of heavy oil combustion [63]. Mehta et al. believe that Co originates from nonferrous or ferrous industries [40]. Furthermore, Ni and Co are widely used in the smelting of ores, alloy manufacturing, and steel production [16]. Chai et al. also found the Co and Ni in surface soil could originate from metal plating and treating industries in industrial areas [25]. Additionally, the average concentrations of Ni and Co in DD were relatively higher than in other kinds of fugitive dust. As the study area is a typical mining and metallurgy area in central China with a long history, the Ni and Co could have been released from different kinds of factories or plants into the atmosphere and deposited on the surface again [67].

The mining-related activities factor profile included high loading values for As, Sn, Zn, and Cr, which were 73.4%, 79.5%, 54.5%, and 42.8%, respectively (Appendix A). Wu et al. revealed that As, Sn, Zn, and Cr are widely used in mining activities and metallurgical processes [68]. Liang et al. found that mining activities can significantly increase the accumulation of As [60], while an increase in Cr is generally believed to result from fossil fuel combustion, which is the main power source for metallurgy [61]. Hu et al. [28] revealed that refining and metallurgical smelting processes release Cr into the atmosphere and soil environment. Moreover, Sn and Zn are released during the mining processes of tin ore and lead-zinc ore, respectively, which are the main metal minerals in the area [64]. Additionally, the average concentrations of these HMs in DD and LD were relatively higher than in other types of fugitive dust (Table 1). The results also indicated that these HMs exist in the local atmosphere and in tailing wastes, which are related to mining activities [40].

Cd and Pb were predominant in the fourth source, and the loading values of these HMs were 79.5% and 82.4%, respectively (Appendix A). The average concentrations of Pb and Cd significantly exceeded the local background values. The results indicated that the accumulation of Cd and Pb in the local fugitive dust could be influenced by anthropogenic activities [58]. Many previous studies have revealed that Pb and Cd are the characteristic HMs of transportation emissions. [69,70,71]. As leaded gasoline is the main energy source for trucks in the area, the combustion of leaded gasoline is an important source of Pb [13]. Moreover, Niu et al. revealed that automobile exhausts could cause the emission of Cd [56]. In addition, Cd is also one of the important materials in tire manufacturing, indicating that the wearing of tires could also increase the concentration of Cd in fugitive dust [71]. Therefore, fuel exhausts, tire wearing, and abrasion of vehicle parts could be responsible for the emission of Pb and Cd.

The contributions of the different sources calculated by the PMF model indicated that the geological background and mineralizing and weathering process were the mainly influences in the accumulation of HMs in the area. The main routes of transportation of HMs from minerals into air, water, and soil were human activities, such as metallurgy, mining, and other industrial activities with a long history [12].

### 3.5. Comparison of Different Source Identification Methods

The results of the source identification obtained from the PCA and PMF models were similar. Compared to the PMF model, the analysis procedure of the PCA model is considered to be relatively simple by some researchers [72]. Sometimes, the PCA model can have relatively low specificity and cannot identify the characteristic markers for various sources [36]. In comparison to other research, Anaman et al. found, through co-application of PCA and PMF models in a smelting site, that grassland received HMs from atmospheric deposition and surface runoff, while forestland soils only received HMs from atmospheric deposition [22]. In the present research, the PCA model could only identify three sources, while the PMF model could classify four factors. The results indicate that PCA explained the dataset through the mixing of sources [73]. Additionally, the percentages of the contributions from each source identified in the PCA and PMF models showed differences. The natural source contribution revealed with PCA was 49.62%, which was relatively higher than that determined with PMF (38.55%). The traffic-related source contribution found with PCA (9.96%) was less than that with PMF (12.06%). Additionally, Gupta et al. also found that the very different percentage contributions of factors in PM_10_ in Delhi could be obtained by using PCA and PMF models [36]. The results indicate the striking differences in the operating methods of PCA and PMF [73].

Generally, the PMF model could substitute missing values and weigh the uncertainty of all samples [68]. However, by integrating correlation analysis, the use of PCA and PMF models in source identification could clearly describe the sources for the accumulation of HMs in fugitive dust. The correlation analysis and PCA model are powerful statistical tools, and they could effectively help the PMF model in determining the source factors and contributions. Firstly, the correlation analysis determined and clarified a strong relationship between different HMs. Then, the PCA model revealed the primary information about different sources. Finally, the PMF model indicated the detailed contributions of different sources [22]. Therefore, the co-application of different source identification methods is vital to understand the sources of HMs.

## 4. Conclusions

Our study comprehensively demonstrated the concentrations and pollution characteristics and identified the sources of HMs in five types of fugitive dust from a typical mining and metallurgy area in Southeast Hubei through the co-application of PCA and PMF methods. Except for Fe and Mn, the results for the average concentrations of Cu, Cd, Pb, As, Zn, Sn, Co, Cr, and Ni were higher than the local background values. The results revealed that the HMs were more likely to accumulate in SD, DD, RD, and LD than in CD. The combined analysis method indicated that traffic-related activities are likely responsible for the accumulation of Pb and Cd, while Fe, Mn, and Cu mainly originate from nature. The accumulation of Ni and Co could be affected by local atmospheric deposition, and the As, Sn, Zn, and Cr might originate from mining-related activities. Moreover, the PMF model also found that natural sources, local deposition, and mining-related activities had relatively high contributions, which were 38.55%, 29.27%, and 20.11%, respectively. The results indicate that the local geological background, mining, metallurgy, and other mining-related industrial activities are mainly responsible for the accumulation of HMs in the area. Additionally, all the results showed that the PCA and PMF methods could complement each other in the identification of the sources of HMs in fugitive dust in the study area. The use of multiple source apportionment models can help governments make targeted control strategies for HM dispersion. However, if governments want to further strengthen the accuracy of source identification in specific fugitive dust or particulate matter classes, more research could still be considered in the future based on fingerprint materials or more influential factors, such as functional zone distribution, land-use types, metallurgy area distributions, and so on.

## Figures and Tables

**Figure 1 ijerph-19-13227-f001:**
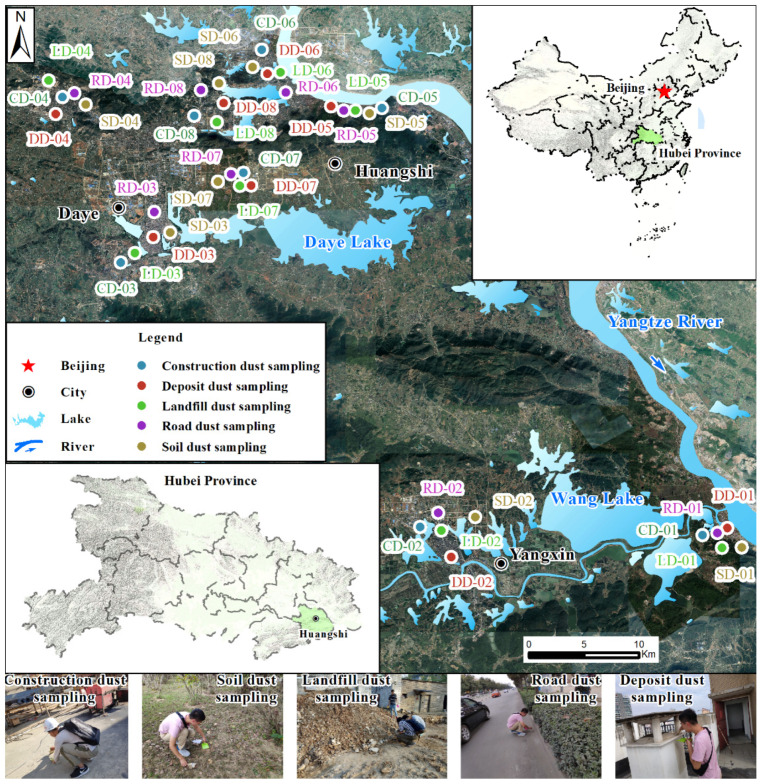
General map showing location of study area and sampling sites.

**Figure 2 ijerph-19-13227-f002:**
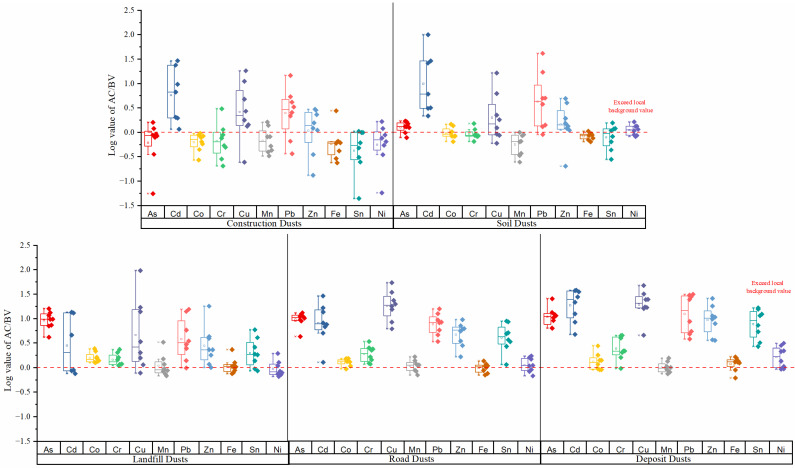
Log values for average accumulation concentration (AC) versus the local background value (BV) of HMs in five different kinds of fugitive dust.

**Figure 3 ijerph-19-13227-f003:**
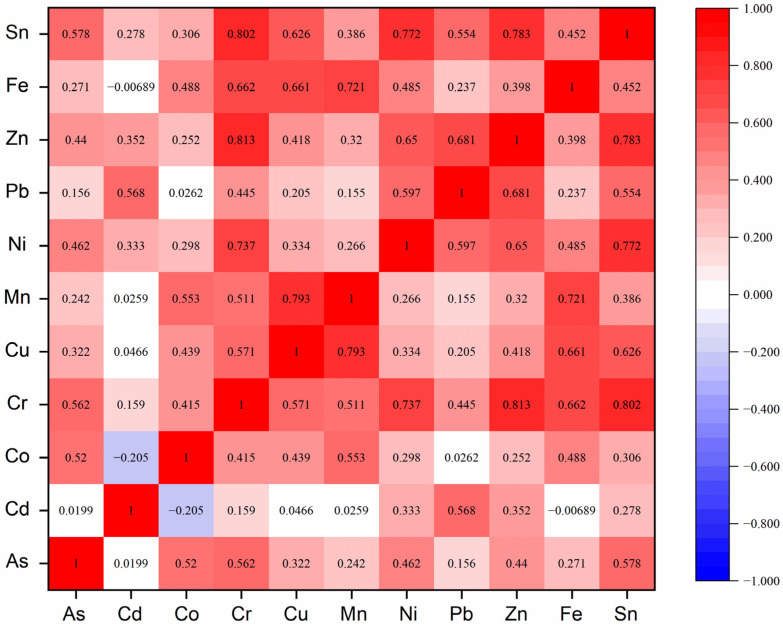
Plot of Pearson correlations of heavy metals in fugitive dust.

**Figure 4 ijerph-19-13227-f004:**
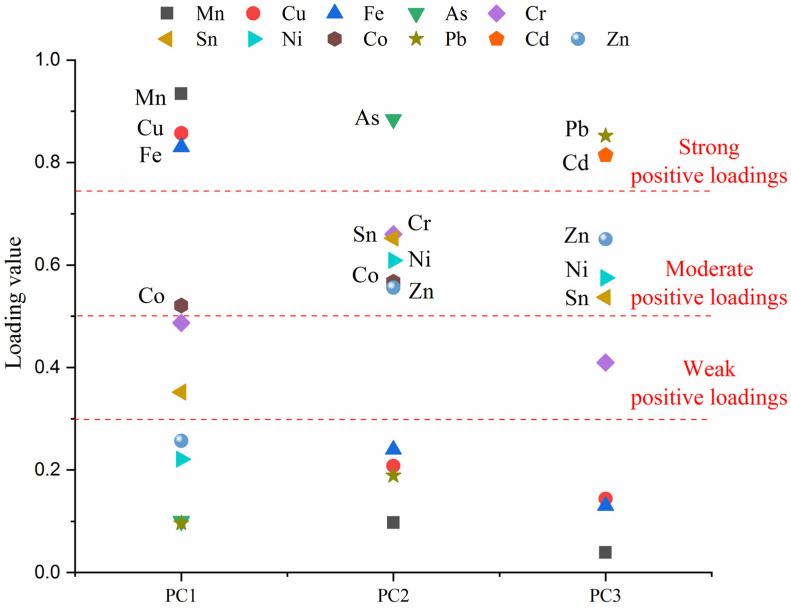
Loadings of HMs on the varimax rotated factor in fugitive dust.

**Figure 5 ijerph-19-13227-f005:**
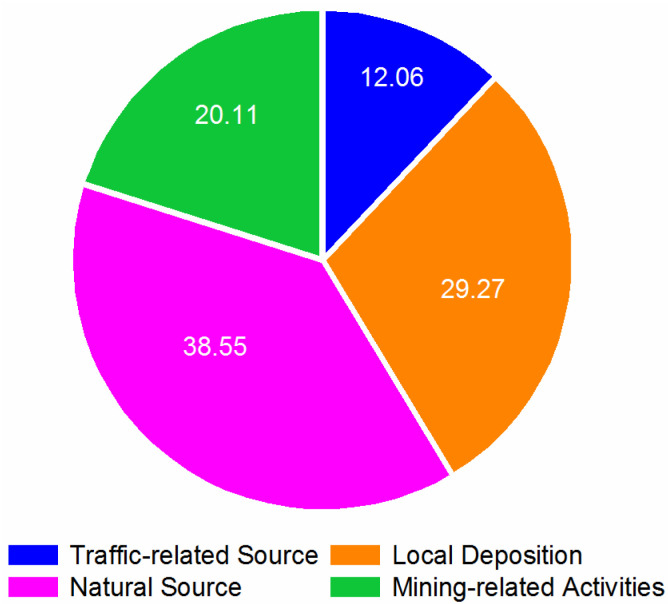
Contributions (%) of the identified sources in fugitive dust.

**Table 1 ijerph-19-13227-t001:** Statistical results for heavy metals in fugitive dusts (unit: mg kg^−1^).

	As	Cd	Co	Cr	Cu	Mn	Ni	Pb	Zn	Fe	Sn
All (N = 40)											
Min	0.680	N.D.	4.16	17.6	7.49	175	2.15	9.71	11.0	9326	0.0967
Max	313	17.1	42.1	387	2962	2340	116	1107	2148	108,302	36.2
Mean ± SD	82.9 ± 69.9	2.36 ± 3.08	18.4 ± 7.56	139 ± 89.5	437 ± 565	683 ± 367	43.8 ± 22.5	240 ± 258	409 ± 450	39,023 ± 18,553	7.88 ± 9.11
CV	84.3%	130%	41.0%	64.4%	129%	53.7%	51.5%	107%	110%	47.5%	116%
CD (N = 8)											
Min	0.680	N.D.	4.16	17.6	7.49	232	2.15	9.71	11.0	9326	0.0967
Max	19.7	5.03	14.6	260	557	1152	62.3	392	248	108,302	2.30
Mean ± SD	10.2 ± 5.45	1.60 ± 1.80	10.4 ± 3.40	78.8 ± 73.0	159 ± 180	561 ± 321	28.3 ± 17.4	111 ± 114	130 ± 83.9	30,304 ± 30,056	1.30 ± 0.793
CV	53.5%	102%	34.7%	91.4%	106%	54.7%	64.1%	107%	65.0%	103%	65.7%
SD (N = 8)											
Min	9.58	N.D.	10.0	55.6	18.2	175	30.4	24.1	16.9	25,023	0.611
Max	21.1	17.1	22.6	129	505	713	60.7	1107	413	41,381	3.45
Mean ± SD	16.1 ± 3.63	3.63 ± 5.39	15.4 ± 4.09	81.4 ± 21.1	116 ± 156	445 ± 189	42.3 ± 9.93	253 ± 348	170 ± 125	32,910 ± 4928	1.97 ± 0.892
CV	22.6%	149%	26.6%	25.9%	134%	42.5%	23.5%	137%	74.0%	15.0%	45.2%
DD (N = 8)											
Min	77.2	0.804	13.7	81.3	140	527	34.4	100	295	23,733	5.84
Max	313	6.56	42.1	387	1455	1099	116	837	2148	64,167	36.2
Mean ± SD	145 ± 70.1	3.98 ± 2.16	21.5 ± 8.96	234 ± 110	700 ± 366	731 ± 185	67.2 ± 30.1	455 ± 310	942 ± 579	48,102 ± 12,416	20.3 ± 11.1
CV	48.5%	54.3%	41.6%	47.0%	52.3%	25.4%	44.8%	68.2%	61.5%	25.8%	54.6%
RD (N = 8)											
Min	52.1	0.218	14.3	100	187	499	25.1	89.8	137	27,311	2.51
Max	160	4.98	23.4	289	1667	1168	63.8	420	791	52,816	19.5
Mean ± SD	122 ± 31.0	1.88 ± 1.40	19.9 ± 2.93	171 ± 61.0	688 ± 448	786 ± 203	43.6 ± 13.2	226 ± 103	440 ± 198	38,922 ± 7962	10.5 ± 6.60
CV	25.4%	74.2%	14.7%	35.6%	65.1%	25.8%	30.4%	45.4%	45.1%	20.5%	53.4%
LD (N = 8)											
Min	51.3	N.D.	19.4	94.2	23.7	482	24.8	26.2	84.1	29,313	1.90
Max	197	2.31	37.3	200	2962	2340	72.5	412	1497	91,309	13.0
Mean ± SD	121 ± 43.4	0.721 ± 0.927	20.0 ± 6.25	129 ± 36.9	522 ± 940	891 ± 575	37.5 ± 15.0	155 ± 142	362 ± 439	44,874 ± 18,595	5.35 ± 3.59
CV	35.8%	129%	25.0%	28.6%	180%	64.5%	40.1%	91.7%	121%	41.4%	67.1%
BV	12.3	0.172	15.4	86	30.7	712	37.3	26.7	83.6	39,100	2.2

N, number of samples; BV, local soil background values (CNEMC, 1990); SD, standard deviation; CV, coefficient of variation; N.D., not detected.

## Data Availability

The data used to support the findings of this study are available from the corresponding author.

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
