# Peer review of "The Predominant Sources of Heavy Metals in Different Types of Fugitive Dust Determined by Principal Component Analysis (PCA) and Positive Matrix Factorization (PMF) Modeling in Southeast Hubei: A Typical Mining and Metallurgy Area in Central China"

_ijerph, 2022, doi:10.3390/ijerph192013227_

Round 1
Reviewer 1 Report
Line 140 Xij schould be explained
The AC - schould be explained
If the AC will be explain the discused then presented graphs will be more clear.
In my opinion the aerosols should be explained specially dust suspended in the air. The coefficent of PM1, PM10, PM100 in air in tested area should be have shown.
The article will be more interesting if the graphs will be more clear and better discused.
Line 224 Pearson correlation - should be explain earlier - why it is so important, then this important graph will be understand for readers.
Author Response
Point 1: Line 140 Xij schould be explained
Response 1: Thank you very much for your good suggestion. We explained the Xij. Revised portions are marked yellow and shown in Line 173.
Point 2: The AC - schould be explained
Response 2: According to the reviewer’s good suggestion, we added the explanation of AC. We changed ‘the results showed the average accumulation concentration versus the local background value (AC/BV) of the heavy metals’ into ‘the results showed the average accumulation concentration (AC) versus the local background value (BV) of the heavy metals’ Revised portions are marked yellow and shown in Line 222-223.
Point 3: If the AC will be explain the discused then presented graphs will be more clear.
Response 3: We appreciate your warm work earnestly. According to the reviewer’s suggestion, we added the explanation of AC. Revised portions are marked yellow and shown in Line 222. Besides that, we did a more detailed analysis of the results of the log value of AC/BV of HMs in five different kinds of fugitive dust (Fig. 2). Revised portions are marked yellow and shown in Line 227-230 and Line 234-248.
Point 4: In my opinion the aerosols should be explained specially dust suspended in the air. The coefficent of PM1, PM10, PM100 in air in tested area should be have shown.
Response 4: We appreciate your warm work earnestly. We agree with the your suggestion, that the research about the source identification of HMs in aerosols, especially in the dust suspended in the air is important. However, fugitive dust is also believed as a major fraction of particulate matter [1]. In recent years, much research about the source identification of pollutants in the different types of fugitive dust, such as road dust [2], construction dust [3], soil dust [4], landfill dust [5], deposition dust [6], and so on, have been done. The results indicate the research source identification of pollutants in fugitive dust is also important for air pollution control. However, most studies always focused on one or two types of fugitive dust. And very few go further to do the source identification with combined source analysis methods in the mining and metallurgy area. In the present study, source identification of HMs in five types of fugitive dust has been done in Southeast Hubei, which is the biggest polymetallic deposit in Central China. Considering the local long-term extensive mining activities, developing characteristics, and green ecological mining transformation strategy, the source identification of HMs in the soil dust, landfill dust, road dust, construction dust, and deposit dust would mainly reflect the transmission path of HMs in the local fugitive dust. The achievements would provide scientific support for making atmospheric pollution accurate control policies. Besides that, we will also do more work on the aerosols such as PM1, PM10, and PM100 in the air in the area. The results will be shown in further research.
- Cao, J.; Shen, Z.; Chow, J.; Watson, J.; Lee, S.; Tie, X.; Ho, K.; Wang, G.; Han, Y. Winter and summer PM2.5 chemical compositions in fourteen Chinese cities. J. Air & Waste Manage. Assoc. 2012, 62:1214-1226.
- Li, T.; Dong, W.; Dai, Q.; Feng, Y.; Bi, X.; Zhang, Y.; Wu, J. Application and validation of the fugitive dust source emission inventory compilation method in Xiong'an New Area, China. Sci. Total Environ. 2021. 798:149114.
- Liu, Y.; Shao, L.; Wang, W.; Chen, J.; Zhang, H.; Yang, Y.; Hu, B. Study on fugitive dust control technologies of agricultural harvesting machinery. Agric. 2022. 12(7):1038.
- Cui, M.; Lu, H.; Etyemezian, V.; Su, Q. Quantifying the emission potentials of fugitive dust sources in Nanjing, East China. Atmos. Environ. 2019. 207:129-135.
- Yen, P.; Chen, W.; Yuan, C.; Tseng, Y.; Lee, J.; Wu, C. Exploratory investigation on the suppression efficiency of fugitive dust emitted from coal stockpile: Comparison of innovative atomizing and traditional spraying technologies. Process Saf. Environ. Prot. 2021. 154:348-359.
- Wang, S.; Hu, G.; Yan, Y.; Wang, S.; Yu, R.; Cui, J. Source apportionment of metal elements in PM2.5 in a coastal city in Southeast China: Combined Pb-Sr-Nd isotopes with PMF method. Atmos. Environ. 2018. 198, 302-312.
Point 5: The article will be more interesting if the graphs will be more clear and better discused.
Response 5: Thank you very much for your good suggestion. We have done a more detailed analysis of all the graphs to make the results better to understand and reflect the main content of the results. Revised portions are marked yellow and shown in ‘Section 3.1’, ‘Section 3.2’, ‘Section 3.3’, ‘Section 3.4’, and ‘Section 3.5’.
Point 6: Line 224 Pearson correlation - should be explain earlier - why it is so important, then this important graph will be understand for readers.
Response 6: Thank you very much for your good suggestion. We have explained the importance of Pearson correlation at beginning of the section. Revised portions are marked yellow and shown in Line 253-259.

Reviewer 2 Report
Summary assessment: The manuscript should be re-edited. The text requires a solid linguistic correction. There are many errors of meaning, terminology and common mistakes.
1. The section Introduction. The meaning of single sentences, and even large fragments of text are incomprehensible. The purpose and scope of the research should be better specified.
2. Verse 99 - “dust” is uncountable noun, “40 dust” sounds bad.
3. Subsection 2.2 Sampling and preparation - Please, explain how it was possible to take dust samples of “at least seven consecutive days” in the only 6-day period (October 15th to 20th, 2020).Please, explain how the samples were dried, and which fraction was taken to chemical examination.
4. Describe in more detail from which surfaces and how the samples were taken marked as CD, SD, LD, RD, DD. Why were dust samplers not used?
5. Why was the deposition examined and not particulate matter? Particulate matter is considered to be much more dangerous to human and animal health.
6. Verse 116 – I suggest change “Eleven kinds of heavy metals” for “Eleven heavy metals”.
7. Verse 121-122: Please, explain what the statement means: “all excellent pure levels of the reagents were used”.
8. 131-132 – I do not understand the sense of the sentence: “It could make it easier to indicate a given multidimensional system by displaying the correlations among the original variables [28]”.
9. 2.4 Principal component analysis - This subsection contains general statements, but does not describe the initial assumptions made. Please complete.
10. 2.5 Positive matrix factorization - This subsection contains theoretical formulas, but does not describe the purpose of the analysis and the initial assumptions made. Please complete.
11. Verse 164 - please, change “the results of HMs in fugitive dust” for “the results of HMs content in fugitive dust”.
12. Table 1 - Please provide a formula for CV. Usually, CV is the ratio SD/Mean. However, the calculated CVV shows that a different formula (what?) was used.
13. If the CV value was incorrectly calculated, the calculations should be corrected and the entire description of the subsection changed, including Figure 2.
14. Subsection 3.2. Correlations of heavy metals in fugitive dust - Conclusions regarding the origin of dust are very poorly documented. This article excerpt should be more persuasive. It should be emphasized that there is no rigid value for the r that would indicate a strong correlation. It all depends on the number of cases. For a small number of cases, the corresponding r values should be particularly high.
15. Figure 4 – Please, enter PC3 on the drawing axis.
16. Figure 5 (and other places) - What does "local despite" mean?
17. Verse 253 - Change Fig. S2 for (probably) Figure 5.
18. Verses 258-259 – “Such as, Pb could be released when the tire and brake were worn” - This is a risky statement with little confirmation in the literature. Why only Pb?
19. Figure 5 – What is on the vertical axes? concentrations (or contents).
20. Verse 272 – You should change the sense “Co has always believed in emissions …”.
21. Versus 293-295 – “Sn is one of the most important rare metal minerals, while Zn is the most important associated mineral in the area” - neither Sn nor Zn are minerals.
22. Verse 303 - In Fig. 5, not S2.
Author Response
Point 1: Summary assessment: The manuscript should be re-edited. The text requires a solid linguistic correction. There are many errors of meaning, terminology and common mistakes.
Response 1: According to the reviewer’s good suggestion, the manuscript was polished very carefully again to make the paper better. Revised portions are marked yellow and ‘Section 1’, ‘Section 2’, ‘Section 3’, and ‘Section 4’.
Point 2. The section Introduction. The meaning of single sentences, and even large fragments of text are incomprehensible. The purpose and scope of the research should be better specified.
Response 2: We appreciate your warm work earnestly. According to the reviewer’s suggestion, we reorganized the ‘Introduction’ section to make it better to understand and reflect the purpose and scope of the research. Revised portions are marked yellow.
Point 3. Verse 99 - “dust” is uncountable noun, “40 dust” sounds bad.
Response 3: Thank you very much for your good suggestion. We changed ‘a total of 40 fugitive dust were sampled’ into ‘a total of 40 fugitive dust samples were collected’. Revised portions are marked yellow and shown in Verse 107-108.
Point 4. Subsection 2.2 Sampling and preparation - Please, explain how it was possible to take dust samples of “at least seven consecutive days” in the only 6-day period (October 15th to 20th, 2020).Please, explain how the samples were dried, and which fraction was taken to chemical examination.
Response 4: We are sorry for confusing you. We tried to express we had collected the fugitive dust samples from October 15th to 20th, 2020. And the sampling work began after at least seven consecutive days without rain. We reorganized the sentence to make it better to understand. Revised portions are marked yellow and shown in Verse 107-110.
Point 5. Describe in more detail from which surfaces and how the samples were taken marked as CD, SD, LD, RD, DD. Why were dust samplers not used?
Response 5: According to the reviewer’s suggestion, we presented the details of the sampling method. Revised portions are marked yellow and shown in Verse 112-122. And we added the sampling photos in Fig. 1 to make it better to understand the sampling method. Revised portions are marked yellow and shown in Verse 105. Moreover, previous studies have already revealed the sampling method used in our research is widely used in research about fugitive dust [1-6]. Besides that, we also agree with the reviewer’s suggestion. More samples will be collected by the dust sampler to do further research on the source identification of HMs in the area. The results will be shown in further research.
- Zhang, Q.; Shen, Z.; Cao, J.; Ho, K.; Zhang, R.; Bie, Z.; Chang, H.; Liu, S. Chemical profiles of urban fugitive dust over Xi’an in the south margin of the Loess Plateau, China. Atmos. Pollut. Res. 2014. 5:421-430.
- Liu, Y.; Wang, X.; Guo, Y.; Mao, Y.; Li, H. Association of black carbon with heavy metals and magnetic properties in soils adjacent to a cement plant, Xuzhou (China). J. Appl. Geophys. 2019. 170,103802.
- Zhong, P.; Zhang, J.; Xu, J.; Tian, Q.; Hu, T.; Gong, X.; Zhan, C.; Liu, S.; Xing, X.; Qi, S. Contamination characteristics of heavy metals in particle size fractions from street dust from an industrial city, Central China. Air Qual. Atmos. Hlth. 2020, 13(7):871-883.
- Liu, S.; Zhang, X.; Zhan, C.; Zhang, J.; Xu, J.; Wang, A.; Zhang, H.; Xu, J.; Guo, J.; Liu, X.; Xing, X.; Cao, J.; Xiao, Y. Evaluating heavy metals contamination in campus dust in Wuhan, the university cluster in Central China: Distribution and potential human health risk analysis. Environ. Earth. Sci. 2022, 81(7):210.
- Chen, H.; Zhan, C.; Liu, S.; Zhang, J.; Liu, H.; Liu, Z.; Liu, T.; Liu, X.; Xiao, W. Pollution Characteristics and Human Health Risk Assessment of Heavy Metals in Street Dust from a Typical Industrial Zone in Wuhan City, Central China. Int. J. Environ. Res. Public Health. 2022, 19, 10970.
- Rahman, M. S.; Khan, M. D. H.; Jolly, Y. N.; Kabir, J.; Akter, S.; Salam, A. Assessing risk to human health for heavy metal contamination through street dust in the Southeast Asian Megacity: Dhaka, Bangladesh. Sci. Total Environ. 2019. 660:1610-1622.
Point 6. Why was the deposition examined and not particulate matter? Particulate matter is considered to be much more dangerous to human and animal health.
Response 6: We appreciate your warm work earnestly. We agree with the reviewer that particulate matter is considered to have potential cancer risks to harm human and animal health. The research on source identification of HMs in particulate matter, such as PM1, PM2.5, PM10, and so on is also important. However, fugitive dust is generally believed as the main fraction of particulate matter [1]. The previous studies had already revealed fugitive dust would account for 12 – 34% of particulate matter in most Chinese cities [2]. In recent years, many studies about the source identification of pollutants in the different types of fugitive dust, such as road dust [3], construction dust [4], soil dust [5], landfill dust [6], deposition dust [7], and so on, had been done. The results indicate the research on source identification of pollutants in fugitive dust is also important for air pollution control. However, most studies always focused on one or two types of fugitive dust. And very few go further to do the source identification with combined source analysis methods in the mining and metallurgy area. In the present study, source identification of HMs in five types of fugitive dust has been done in Southeast Hubei, which is the biggest polymetallic deposit in Central China. Considering the local long-term extensive mining activities, developing characteristics, and green ecological mining transformation strategy, the source identification of HMs in the soil dust, landfill dust, road dust, construction dust, and deposit dust would mainly reflect the transmission path of HMs in the local fugitive dust. The achievements would provide scientific support for making atmospheric pollution accurate control policies. Besides that, we will also do more work on the particulate matter in the area as you suggested. The results will be shown in further research.
- Kolesar, K. R.; Schaaf, M. D.; Bannister, J. W.; Schreuder, M. D.; Heilmann, M. H. Characterization of potential fugitive dust emissions within the Keeler Dunes, an inland dune field in the Owens Valley, California, United States. Aeolian. Res. 2022, 54:100765.
- Cao, J.; Shen, Z.; Chow, J.; Watson, J.; Lee, S.; Tie, X.; Ho, K.; Wang, G.; Han, Y. Winter and summer PM2.5 chemical compositions in fourteen Chinese cities. J. Air & Waste Manage. Assoc. 2012, 62:1214-1226.
- Li, T.; Dong, W.; Dai, Q.; Feng, Y.; Bi, X.; Zhang, Y.; Wu, J. Application and validation of the fugitive dust source emission inventory compilation method in Xiong'an New Area, China. Sci. Total Environ. 2021. 798:149114.
- Liu, Y.; Shao, L.; Wang, W.; Chen, J.; Zhang, H.; Yang, Y.; Hu, B. Study on fugitive dust control technologies of agricultural harvesting machinery. Agric. 2022. 12(7):1038.
- Cui, M.; Lu, H.; Etyemezian, V.; Su, Q. Quantifying the emission potentials of fugitive dust sources in Nanjing, East China. Atmos. Environ. 2019. 207:129-135.
- Yen, P.; Chen, W.; Yuan, C.; Tseng, Y.; Lee, J.; Wu, C. Exploratory investigation on the suppression efficiency of fugitive dust emitted from coal stockpile: Comparison of innovative atomizing and traditional spraying technologies. Process Saf. Environ. Prot. 2021. 154:348-359.
- Zhang, Q.; Shen, Z.; Cao, J.; Ho, K.; Zhang, R.; Bie, Z.; Chang, H.; Liu, S. Chemical profiles of urban fugitive dust over Xi’an in the south margin of the Loess Plateau, China. Atmos. Pollut. Res. 2014. 5:421-430.
Point 7. Verse 116 – I suggest change “Eleven kinds of heavy metals” for “Eleven heavy metals”.
Response 7: We appreciate your warm work earnestly. According to the reviewer’s suggestion, we changed the ‘Eleven kinds of heavy metals’ into ‘Eleven heavy metals’. Revised portions are marked yellow and shown in Verse 131.
Point 8. Verse 121-122: Please, explain what the statement means: “all excellent pure levels of the reagents were used”.
Response 8: We are sorry for the inappropriate statement. We tried to express all the reagents, which we used for the chemical analysis were guaranteed reagents to avoid contamination of the samples [1]. The guaranteed reagents with high purity grades were generally used for sophisticated chemical analysis [2]. We re-edited the sentence. Revised portions are marked yellow and shown in Verse 136-137.
- Zhong, P.; Zhang, J.; Xu, J.; Tian, Q.; Hu, T.; Gong, X.; Zhan, C.; Liu, S.; Xing, X.; Qi, S. Contamination characteristics of heavy metals in particle size fractions from street dust from an industrial city, Central China. Air Qual., Atmos. Health. 2020, 13(7):871-883.
- Du, Y.; Zhou, H.; Ju, X.; Hao, Hong.; Yin S. Health risk assessment of heavy metals in road dusts in urban parks of Beijing, China. Procedia Environ. Sci. 2013, 18:299-309.
Point 9. 131-132 – I do not understand the sense of the sentence: “It could make it easier to indicate a given multidimensional system by displaying the correlations among the original variables [28]”.
Response 9: We appreciate your warm work earnestly. We also found the section wasn’t well expressed. We removed the sentence and re-edited the section to make it better to understand. Revised portions are marked yellow and shown in Verse 145-162.
Point 10. 2.4 Principal component analysis - This subsection contains general statements, but does not describe the initial assumptions made. Please complete.
Response 10: Thank you very much for your good suggestion. We re-edited this section and added the necessary contents as the reviewer’s suggestion. Revised portions are marked yellow and shown in Verse 145-162.
Point 11. 2.5 Positive matrix factorization - This subsection contains theoretical formulas, but does not describe the purpose of the analysis and the initial assumptions made. Please complete.
Response 11: Thank you very much for your good suggestion. We added the purpose of the analysis and the initial assumptions as the reviewer’s suggestion. Revised portions are marked yellow and shown in Verse 164-170.
Point 12. Verse 164 - please, change “the results of HMs in fugitive dust” for “the results of HMs content in fugitive dust”.
Response 12: We appreciate your warm work earnestly. According to the reviewer’s suggestion, we changed the ‘the results of HMs in fugitive dust’ into ‘the results of HMs content in fugitive dust’. Revised portions are marked yellow and shown in Verse 197.
Point 13. Table 1 - Please provide a formula for CV. Usually, CV is the ratio SD/Mean. However, the calculated CVV shows that a different formula (what?) was used.
Response 13: We appreciate your warm work earnestly. We feel sorry about the wrong formula to calculate the CV value. It should be calculated with the ratio SD/Mean. We re-calculated the CV value. Revised portions are marked yellow and shown in Table 1.
Point 14. If the CV value was incorrectly calculated, the calculations should be corrected and the entire description of the subsection changed, including Figure 2.
Response 14: We appreciate your warm work earnestly. We re-calculated the CV value with the ratio SD/Mean. Additionally, we also re-edited the ‘3.1 section’ with new results and do more work on the analysis of Figure 2. Revised portions are marked yellow and shown in Verse 202-248.
Point 15. Subsection 3.2. Correlations of heavy metals in fugitive dust - Conclusions regarding the origin of dust are very poorly documented. This article excerpt should be more persuasive. It should be emphasized that there is no rigid value for the r that would indicate a strong correlation. It all depends on the number of cases. For a small number of cases, the corresponding r values should be particularly high.
Response 15: Thank you very much for your good suggestion. We have to do more work on the analysis of the correlation results of heavy metals in fugitive dust. Revised portions are marked yellow and shown in the ‘3.2 section’.
Point 16. Figure 4 – Please, enter PC3 on the drawing axis.
Response 16: We appreciate your warm work earnestly. And we are sorry for the mistakes in the drawing. We updated Figure 4. Revised portions are marked yellow and shown in Figure 4.
Point 17. Figure 5 (and other places) - What does "local despite" mean?
Response 17: We appreciate your warm work earnestly. We feel sorry about the spelling mistakes. We tried to express local deposition. We re-drawed the graph and updated other places in the manuscript. Revised portions are marked yellow and shown in the manuscript and figures.
Point 18. Verse 253 - Change Fig. S2 for (probably) Figure 5.
Response 18: According to the reviewer’s suggestion, we changed Fig. S2 for Figure 5. Revised portions are marked yellow and shown in Verse 372 in the manuscript and Appendice, respectively. We also updated the figure number in the manuscript and reorganized the ‘Quantitative source apportionment by PMF’ section.
Point 19. Verses 258-259 – “Such as, Pb could be released when the tire and brake were worn” - This is a risky statement with little confirmation in the literature. Why only Pb?
Response 19: Thank you very much for your good suggestion. We also found it was an inappropriate statement. We removed it from the manuscript.
Point 20. Figure 5 – What is on the vertical axes? concentrations (or contents).
Response 20: Thank you very much for your good suggestion. We are sorry for the mistakes in the drawing. We re-checked the graph and the user guide of PMF 5.0. According to the user guide of PMF 5.0, in the graph, the concentration of each species is apportioned to the factor as a pink bar and the percent of each species is apportioned to the factor as blue dots. The concentration bar corresponds to the left y-axis, which is a logarithmic scale. The percent of species corresponds to the right y-axis [1]. We also updated the graph to make it better to understand. Revised portions are marked yellow and shown in Fig. S2 in Appendices.
- USEPA. EPA Positive Matrix Factorization (PMF) 5.0 Fundamentals and User Guide. U.S. Environment Protection Agency, Washington, DC, USA. 2014.
Point 21. Verse 272 – You should change the sense “Co has always believed in emissions …”.
Response 21: Thank you very much for your good suggestion. We changed the ‘And Co has always believed in emissions from nonferrous industries or ferrous industries’ into ‘Mehta et al. believed Co would source from nonferrous industries or ferrous industries’. Revised portions are marked yellow and shown in Verse 339-340.
Point 22. Versus 293-295 – “Sn is one of the most important rare metal minerals, while Zn is the most important associated mineral in the area” - neither Sn nor Zn are minerals.
Response 22: Thank you very much for your good suggestion. We changed the ‘Sn is one of the most important rare metal minerals, while Zn is the most important associated mineral in the area’ into ‘Sn and Zn would be released during the mining process of tin ore and lead-zinc ore, respectively, which are main metal minerals in the area’. Revised portions are marked yellow and shown in Verse 355-356.
Point 23. Verse 303 - In Fig. 5, not S2.
Response 23: Thank you very much for your good suggestion. We changed the figure as you suggested. Revised portions are marked yellow and shown in Verse 323 in the manuscript and Appendice, respectively.

Reviewer 3 Report
The paper presented seems to me to be interesting, current and responds to the social concern that citizens have about the dust they breathe and touch. The treatments of the data by PCA and PMF seem to me to be very appropriate.
The major problem I see with the study is that 40 samples seem to me to be too few samples, because they are few from each of the possible origins, only 8 samples: construction fugitive dust, soil fugitive dust, landfill fugitive dust and deposit fugitive dust.
They seem to me to be too few samples because the coefficients of variation (CV) are very high, in the order of 300 to 600 %, the standard deviation is also very high, too much variability to draw conclusions and trends. That is why I believe that a larger sampling is very necessary.
I reconsider after major revision because of the large variability of the sample samples which I believe does not lead to convincing conclusions. I leave it up to the editors whether or not to publish the paper.
Regarding the format, in the bibliography the name of the journals should be abbreviated in italics but not in capital letters. The volume of the journal should be in italics, which it is not.
Author Response
Point 1: The paper presented seems to me to be interesting, current and responds to the social concern that citizens have about the dust they breathe and touch. The treatments of the data by PCA and PMF seem to me to be very appropriate.
The major problem I see with the study is that 40 samples seem to me to be too few samples, because they are few from each of the possible origins, only 8 samples: construction fugitive dust, soil fugitive dust, landfill fugitive dust and deposit fugitive dust.
They seem to me to be too few samples because the coefficients of variation (CV) are very high, in the order of 300 to 600 %, the standard deviation is also very high, too much variability to draw conclusions and trends. That is why I believe that a larger sampling is very necessary.
I reconsider after major revision because of the large variability of the sample samples which I believe does not lead to convincing conclusions. I leave it up to the editors whether or not to publish the paper.
Response 1: We appreciate your warm work earnestly. We feel sorry about the wrong formula to calculate the CV value. It should be calculated with the ratio SD/Mean. We re-calculated the CV value. Revised portions are marked yellow and shown in Table 1 and line. According to the new results, the coefficient of variation (CV) of HMs was relatively high, which was higher than 41.0%. The results indicated the spatial distribution of HMs would be heterogeneous in the local area [1]. Moreover, in all detected HMs, the CV values of Cd, Cu, Pb, Zn, and Sn were 130%, 129%, 107%, 110%, and 116%, respectively. As the study area is the biggest mining and metallurgy area in Central China, with long-term mining history and the copper ore, tin ore, and lead-zinc ore are the main minerals in the area, the local mining and metallurgy activities, industrial activities, industrial activities, traffic-related activities, and so on would cause such heavy metals heterogeneous emission [2, 3]. Moreover, the CV values of most HMs, such as Cd, Cu, Mn, Pb, Zn, and Sn excessed 54.7% in CD and LD. The results might relate to the heterogeneous distribution of construction sites and landfill sites in the area [4]. Additionally, the emission of HMs in such sites was generally influenced by the construction materials or landfill waste, which might always cause the difference in the HMs concentration from site to site [5]. While the CV values of As, Co, Cr, Mn, and Fe in DD and RD were lower than 48.5%. The results indicated such HMs were relatively homogeneous distribution, which might cause by the local atmospheric deposition [6]. We also re-edited ‘Section 3.1’ to make it better to understand. Besides that, we also agree with the reviewer’s suggestion. We will collect more fugitive dust samples to improve the results of our research. The achievements will be shown in further research.
- Cui, X.; Wang, X.; Liu, B. The characteristics of heavy metal pollution in surface dust in Tangshan, a heavily industrialized city in North China, and an assessment of associated health risks. J. Geochem. Explor. 2020, 210:106432
- Zhong, P.; Zhang, J.; Xu, J.; Tian, Q.; Hu, T.; Gong, X.; Zhan, C.; Liu, S.; Xing, X.; Qi, S. Contamination characteristics of heavy metals in particle size fractions from street dust from an industrial city, Central China. Air Qual., Atmos. Healt. 2020, 13(7):871-883.
- Xu, D.; Zhang, J.; Yan, B.; Liu, H.; Zhang, L.; Zhan, C.; Zhang, L.; Zhong, P. Contamination characteristics and potential envi-ronmental implications of heavy metals in road dusts in typical industrial and agricultural cities, Southeastern Hubei Prov-ince, Central China. Environ. Sci. Pollut. R. 2018, 25:36223-36238.
- Zhang, Q.; Shen, Z.; Cao, J.; Ho, K.; Zhang, R.; Bie, Z.; Chang, H.; Liu, S. Chemical profiles of urban fugitive dust over Xi’an in the south margin of the Loess Plateau, China. Atmos. Pollut. Res. 2014. 5:421-430.
- Mehta, U. H.; Kaul, D, S.; Westerdahl, D.; Ning, Z.; Zhang, K.; Sun, L.; Wei, P.; Gajjar, H, G.; Jeyaraman, J. D.; Patel, M. V.; Joshi, R. R. Understanding the sources of heavy metal pollution in ambient air of neighboring a solid waste landfill site. Aerosol. Sci. Eng. 2022, 6(2):161-175.
- Wu, T.; Bi, X.; Li, Z.; Sun, G.; Feng, X.; Shang, L.; Zhang, H.; He, T.; Chen, J. Contaminations, sources, and health risk of trace metal(-loid)s in street dust of a small city impacted by artisanal Zn smelting activities. Int. J. Env. Res. Pub. He. 2017, 14(9):961.
Point 2: Regarding the format, in the bibliography the name of the journals should be abbreviated in italics but not in capital letters. The volume of the journal should be in italics, which it is not.
Response 2: We appreciate your warm work earnestly. According to the reviewer’s suggestion, we updated the format of all the references. Revised portions are marked yellow and shown in ‘References’.

Round 2
Reviewer 2 Report
I accept the article. Most of my comments were taken into account in the new version of the article. I am asking you to improve a few details:
1. Verse 149 - please, change “transportation” for “transformation”.
2. Verse 271 - Change “manganese is the most important associated mineral” for “manganese is the most important element associated”. Manganese is not a mineral.
3. Verses 327-371 – The discussion in the lines 327-371 should refer to the tables and figures included in the article. Please enter the appropriate sources for the quoted values.
4. Verses 355-356 – “Sn and Zn would be released during the mining process of tin ore and lead-zinc ore, respectively, which are the main metal minerals in the area” - neither Sn nor Zn are minerals, they are elements.
Reviewer 3 Report
I consider that the paper has improved and I would accept it to publish in this form